# Ultrabroadband sound control with deep-subwavelength plasmacoustic metalayers

Stanislav Sergeev [1], Romain Fleury [2] & Hervé Lissek [1] ✉

Controlling audible sound requires inherently broadband and subwavelength acoustic solutions, which are to date, crucially missing. This includes current noise absorption methods, such as porous materials or acoustic resonators, which are typically inefficient below 1 kHz, or fundamentally narrowband. Here, we solve this vexing issue by introducing the concept of plasmacoustic metalayers. We demonstrate that the dynamics of small layers of air plasma can be controlled to interact with sound in an ultrabroadband way and over deep-subwavelength distances. Exploiting the unique physics of plasmacoustic metalayers, we experimentally demonstrate perfect sound absorption and tunable acoustic reflection over two frequency decades, from several Hz to the kHz range, with transparent plasma layers of thicknesses down to $\lambda/1000$. Such bandwidth and compactness are required in a variety of applications, including noise control, audio-engineering, room acoustics, imaging and metamaterial design.

Sound, as a key vector of information and energy, is ubiquitous in our everyday lives. Its importance goes way beyond our daily conversations, and impacts a broad range of applications from the conception of noise-free systems, isolated buildings, to high-quality audio systems, medical therapy, and imaging. Yet, due to the macroscopically large wavelength and the broad frequency range involved, controlling audible sound with structured materials remains to date a difficult challenge[1–3]. Fundamentally, any passive causal acoustic structure is constrained by sum rules that translate into unavoidable bounds between the device size (in units of the acoustic wavelength) and the bandwidth of operation[4]. At low audible frequencies, where the sonic wavelength can be several meters long, acoustic solutions, for example absorbing walls or diffusers, are inherently massive. Devices based on narrow-band resonances, such as locally-resonant metamaterials, can however still be relevant in this regime, at the drastic cost of sacrificing the bandwidth, leading to remarkable but band-limited possibilities for perfect absorption[3,5–9], assymetric transmission[10,11], or wave bending and focusing, among others.

This vexing bottleneck may be avoided by breaking passivity and using active control schemes[12]. Applied to subwavelength acoustic resonators, active approaches have led to solutions with increased reconfigurability and bandwidth[13–17]. Unfortunately, such advances have remained relatively limited by inherent energy and stability constraints related to the inertia of the transducers used to implement them. Even in active schemes, based for example on electrodynamic or piezoelectric transducers, the achievable bandwidth can not be extended at will. This fundamental limitation is related to the increasingly large amount of energy required to actuate resonators far from their resonance condition, which leads to unavoidable instabilities, ultimately restricting the bandwidth.

In this paper, we propose to leverage the inherently non-inertial dynamics of ultrathin layers of air plasma to construct fundamentally broadband active plasmacoustic layers that can manipulate sound over >2 decades, with systems of unprecedentedly small sizes, down to $\lambda/1000$. Our method can manipulate an acoustic field by directly steering fluid particles, without resorting to any non-fluidic interface, simply by leveraging the partial ionization of air, controlled by an electrical field. Our theory of plasmacoustic metalayers unlocks the possibility to use the associated acoustic monopolar and dipolar sources to control the plasmacoustic surface impedance. We experimentally demonstrate practical applications of the concept to extremely thin and broadband perfect sound absorbers and tunable mirrors, with bandwith/size ratios at least 3 orders of magnitude larger than the currently available solutions.

[1]Signal Processing Laboratory LTS2, EPFL, Lausanne, Switzerland. [2]Laboratory of Wave Engineering, EPFL, Lausanne, Switzerland.
✉e-mail: herve.lissek@epfl.ch

# Results

## The subwavelength plasmacoustic metalayer

The principal idea behind the design of the plasmacoustic metalayer is the use of a controlled dynamic corona discharge. This air-ionization phenomenon is used, for example, in flow control[18], and for sound generation[19–21]. The plasmacoustic metalayer considered in this work is schematically illustrated in Fig. 1. It consists of two metallic electrodes separated by an air gap. One electrode (emitter) is represented by a set of thin wires and the second (collector) by a coarse grid. With such design, when no voltage is applied between the electrodes, the structure is acoustically transparent in the audible range. If the system is terminated with a rigid enclosure, an incident sound wave reflects without loss in amplitude (Fig. 1a). A completely different behavior occurs when the unit cell is supplied with power. If a positive constant high voltage is applied to the emitter while the collector is grounded, the magnitude of the electrical field can locally become higher than the breakdown threshold in air, causing an ionization process in a thin region around the emitter wire (violet glow in Fig. 1b). The produced positive ions further drift from the emitter to the collector electrode. Since the energy gained in the electric field in the main volume between the electrodes is not high enough to cause further ionization, the ions interact with surrounding neutral air particles in elastic collisions. This mechanism generates a constant force **F** that pushes the air particles. In the ionization region, a significant amount of the supplied power transforms into heat $H$ through inelastic processes. If the voltage difference across the electrodes varies around a constant value, the fluctuation of the force and heat release can lead to sound generation. The force **F** acts as a dipolar acoustic source, similar to what happens in membrane-based electroacoustic actuators, yet with very small inertia. This dipolar response is indicated by the blue waves propagating with opposite phases and directions in the drift region. The fluctuation of heat has a monopolar influence on the generated acoustic pressure (red waves propagating in phase towards the two opposite directions). Altogether, these monopolar and dipolar acoustic sources have been created without resorting to inertial elements, escaping the inherent drawbacks of resonators, while being able to interact with the sound field in a controlled manner over an arbitrarily small thickness.

Let us now model more accurately these two acoustic sources. The voltage-current curve can be approximated by the Townsend formula[22]: $I = CU(U − U_0)$ which is valid for various corona discharge geometries. Here, $U$ and $I$ are the total discharge voltage and current, $U_0$ is the critical voltage above which the discharge initiates, and $C$ [A/V²] is a dimensional constant that depends on the discharge geometry and gas properties. To generate a sinusoidal sound wave at frequency $\omega$, one should apply a voltage difference in the form $U = U_{DC} + u_{AC} \sin(\omega t)$, with $u_{AC}$ being only a few percents of $U_{DC}$. In the linear approximation, the magnitude of the AC part of the force in the plasmacoustic metalayer with geometry illustrated in Fig. 1 can be expressed as follows:

$$F_\omega = \frac{Cd}{\mu_i}(2U_{DC} − U_0)u_{AC}, \tag{1}$$

where $d$ is the interelectrode distance and $\mu_i$ is the effective ion mobility in the air. The heat power is assumed to be the total Joule losses in the discharge $H(t) \approx U(t)I(t)$, and its AC linear component in the frequency domain can be written as:

$$H_\omega = C\left(3U_{DC}^2 − 2U_{DC}U_0\right)u_{AC}. \tag{2}$$

These two phenomena can be included as sound source terms in the acoustic wave equation. Derivation of equations ((1)–(2)) and their contributions to the total generated pressure in free field are discussed in the previous work[23]. In[24], an attempt was made to control the corona discharge transducer, where it was used as a generic transducer to absorb sound, without benefiting from the physics of the corona discharge. However, equations ((1)–(2)) are frequency-independent, which suggests that they can form the basis for an active actuator that can be controlled over a potentially broad frequency range, as we now demonstrate.

## Active plasmacoustic control

To demonstrate the potential of the plasmacoustic metalayer for sound manipulation, we must identify a control law between an acoustic quantity and the discharge voltage, hence making the plasmacoustic metalayer an actively controlled acoustic device. An interesting target application is perfect sound absorption, which is achieved by matching the metalayer surface acoustic impedance to the characteristic impedance of air $Z_c = \rho_0 c$, where $\rho_0$ is the air density and $c$ is the speed of sound. For this purpose, let us mount the plasmacoustic metalayer right before the termination of an air-filled duct of same cross-section (Fig. 2a). Since the corona discharge is homogeneous along the transverse directions[23] and the interelectrode distance is limited to a few millimeters, force and heat can be considered,

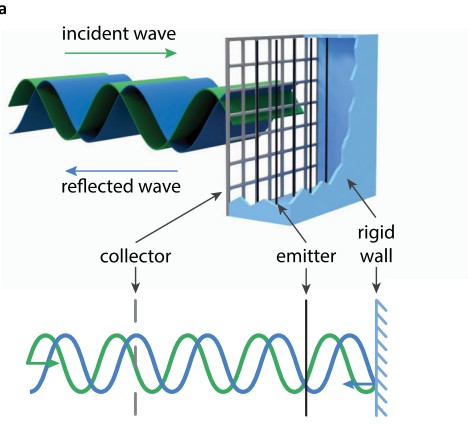
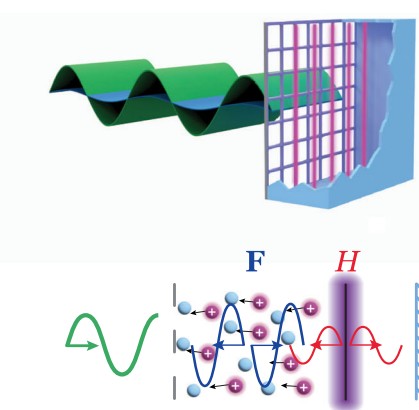

**Fig. 1 | Controlling sound with subwavelength plasmacoustic metalayers.** The metalayer is composed of two electrodes made of thin conducting wires and a grid, spaced by a deep-subwavelength distance. For illustration purposes, the acoustic wavelength is not to scale. **a** Plasmacoustic metalayer placed in front of a rigid wall, when used for controlling acoustic reflection. When no voltage is applied between the collector and the emitter, the metalayer is essentially acoustically transparent. **b** When applying a biased sinusoidal voltage to the electrodes, the induced corona discharge forms a complex acoustic source consisting of a monopolar heat source $H$ centered around the emitter (in red), and a dipolar force source **F** located between the electrodes (in blue). By controlling these two sources, synchronizing them with the incident wave, it is possible to control the total impedance of the wall in a broadband way, for example turning it into a perfect sound absorber.

**a**

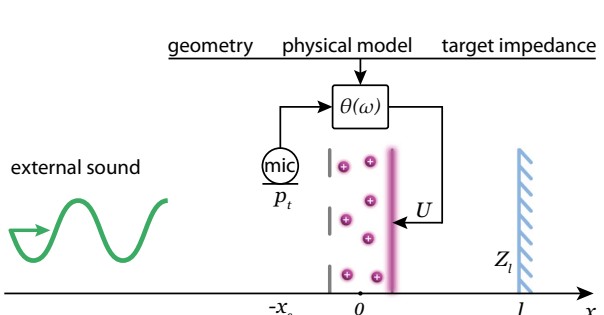

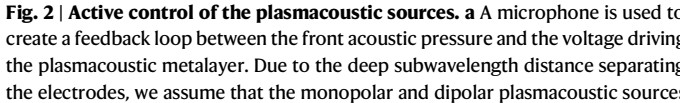

**b**

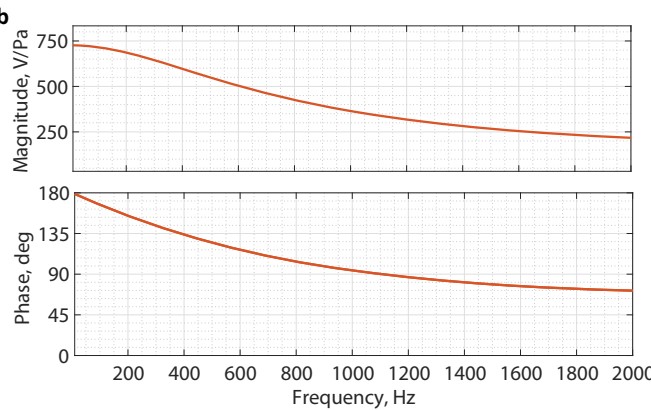

**Fig. 2 | Active control of the plasmacoustic sources. a** A microphone is used to create a feedback loop between the front acoustic pressure and the voltage driving the plasmacoustic metalayer. Due to the deep subwavelength distance separating the electrodes, we assume that the monopolar and dipolar plasmacoustic sources are punctual and both located at the origin. The microphone measures the total pressure $p_t$ at position $-x_0$. The acoustic medium is terminated by a boundary with general acoustic impedance $Z_l$. **b** Bode plot of the control transfer function from the total pressure $p_t$ to the applied AC voltage $u_{AC}$ to target full sound absorption.

at low frequencies, as two collocated acoustic point sources. The termination of the duct is represented by an impedance $Z_l$ at distance $l$ from the center of the metalayer, which has a large real value in the case of a rigid termination. It defines absorption and reflection properties when the metalayer is not actuated (passive regime). The total acoustic pressure in front of the metalayer at position $-x_0$, where we wish to control the acoustic impedance, is sensed by a microphone. When the plasmacoustic metalayer is active, and in the presence of an exogenous acoustic field (characterized by sound pressure $p_{ac}$ and particle velocity $v_{ac}$, accounting for both the incident and reflected sound waves), the total sound pressure $p_t$ and particle velocity $v_t$ at the microphone position $-x_0$ is the sum of the terms produced by all the sources:

$$p_t = p_f + p_h + p_{ac},$$
$$v_t = v_f + v_h + v_{ac}. \qquad (3)$$

In equation (3), $p_f, p_h, v_f$ and $v_h$ are the sound pressures and particle velocities generated by the force and heat sources respectively (see the derivation and breakdown of their contributions to the final system response in Methods). These quantities are directly derived from $F_\omega$ and $H_\omega$ from equations (1) and (2), which depend only on the input voltage $u_{AC}$. In the passive regime, the plasmacoustic sources are equal to zero and the total pressure and velocity are limited to terms produced by the external sound source ($p_{ac}, v_{ac}$). Then, $p_{ac}(-x_0)$ and $v_{ac}(-x_0)$ are linked through the impedance $Z_{ac}$ in a lossless transmission line at distance $x_0 + l$ from the termination with impedance $Z_l$ as:

$$\frac{p_{ac}(-x_0)}{v_{ac}(-x_0)} = Z_{ac} = Z_c \frac{Z_l + jZ_c \tan(k(l+x_0))}{Z_c + jZ_l \tan(k(l+x_0))}. \qquad (4)$$

The same relation holds in the active case since the operation of the metalayer does not introduce any physical interface which could cause scattering of the incident sound wave. Since the control aims at creating an impedance matched condition to absorb sound, the total pressure and velocity at the microphone position should be such that $p_t(-x_0)/v_t(-x_0) = Z_{tg} = Z_c$ with $Z_{tg}$ the target acoustic impedance equal to the characteristic impedance in the air. This equation, together with (3) and (4) allows deriving a control transfer function $\theta(\omega) = u_{AC}(\omega)/p_t(-x_0, \omega)$ driving the plasmacoustic metalayer with the voltage $u_{AC}$ that allows to perfectly absorb the incident sound wave (see Methods). The Bode plot with the magnitude and phase of the control transfer function is shown on panel b in Fig. 2.

## Experimental validation

Our prototype of active plasmacoustic metalayer is illustrated in Fig. 3b. The high voltage electrode is made of 0.1 mm diameter nichrome wire. It is arranged in a back and forth pattern of 5 parallel wire lengths spaced by 10 mm and hold in place by a rigid plastic frame. The wire is strung parallel to the second grounded electrode made of perforated stainless steel plate with rather low flow resistance (2% of characteristic air impedance). The actuator hollow area is $50 \times 50$ mm$^2$ and the distance between the electrodes is 6 mm. When the actuator is biased with a positive voltage of 8 kV, a stable corona discharge is produced with a homogeneous glow along the wire lengths. For sound absorption measurements, the electrodes are enclosed in a rigid plastic termination located 25 mm from the actuator center. We have placed the system in an impedance tube of the same cross-section as the active area (Fig. 3a) and evaluated the absorption performance. The details of the control implementation and the experiment are given in Methods. All necessary parameters for deriving the analytical model of the plasmacoustic metalayer can be estimated by measuring the voltage-current characteristics of the discharge. As expected, when the plasmacoustic material is not controlled, the behavior is governed only by the termination impedance $Z_l$ and leads to the very low sound absorption of about $\alpha_0 = 0.1$, almost constant over the considered frequency range (red dashed line in Fig. 3c). When the actuator is active, with a target impedance equal to the characteristic impedance of air, it is possible to reach a perfect sound absorption over a wide frequency range ($\alpha > 0.98$ between 40 and 1940 Hz, and $\alpha > 0.94$ already from 20 Hz). It is remarkable that with a thickness of only 31 mm, such active absorber is able to effectively address wavelengths up to $\approx 17$ m (note that the frequency scale in Fig. 3c is logarithmic). Such subwavelength acoustic control can theoretically be extended down to even lower frequencies, but limitations of the experimental setup prevent us from measuring lower frequencies with a sufficient signal-to-noise ratio. The highest considered frequency is limited due to several assumptions in the analytical model that do not hold anymore at high frequencies (in particular, the collocation of heat and force sources), and delays in the control system that lead to an unavoidable drift in the achieved impedance[25], as well as the approximation of the control transfer function by rational polynomials of finite order. As a reference case, we also plot the absorption curve of a porous material of similar thickness backed by a hard wall. Clearly, it does not provide sufficient energy dissipation at low frequencies, as illustrated by the blue curve in Fig. 3c. Although passive resonators of comparable size can be designed to absorb low frequency sound, their performance is much more limited frequency-

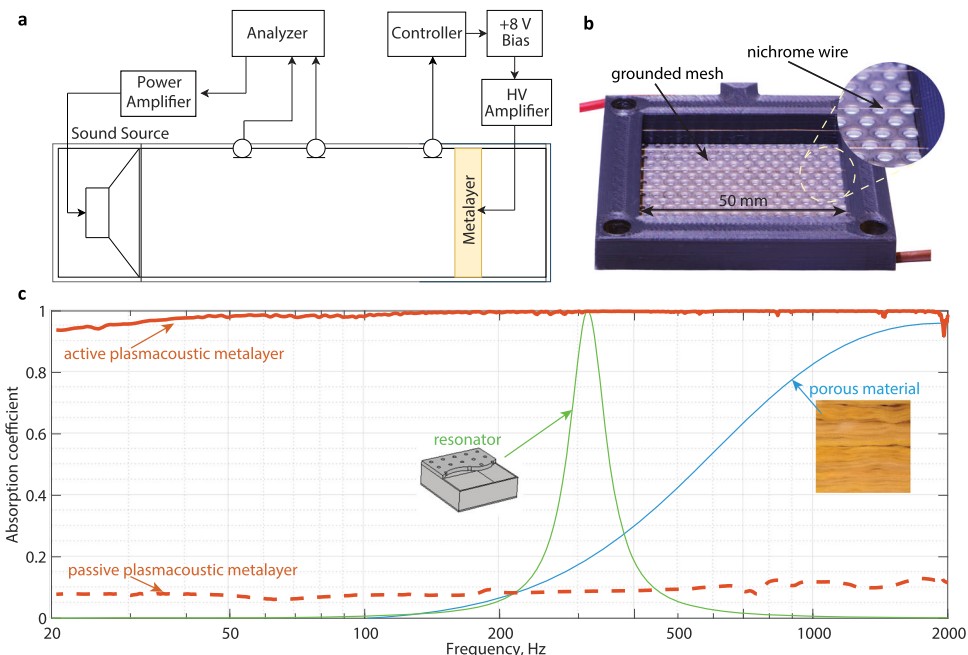

**Fig. 3 | Experimental validation of ultrabroadband absorption from a sub-wavelength plasmacoustic metalayer. a** The set-up is composed of an impedance tube excited from the left and terminated by the active plamacoustic metalayer. **b** Photograph of the prototype, composed of a nichrome wire emitting electrode stretched along parallel lines using a plastic frame, and a grounded grid forming the collector. **c** Measured sound absorption of the active plasmacoustic metalayer from 20 to 2000 Hz, demonstrating its quasi-ideal unitary absorption over two frequency decades (plain red line, the horizontal scale is logarithmic). As references, we also plot the measured absorption of the prototype in the passive case (red dashed line), as well as the theoretical absorption of a porous material layer (blue line) and a resonator (green line) with similar dimensions (the absorption curves are calculated from the analytical models taken from[36]).

wise (see green curve on Fig. 3c). Alternatively, active resonators or metamaterials could be used to extend the low-frequency sound absorption[6,9,14], however their resonant nature prevents from achieving purely real acoustic impedance, leading to band-pass sound absorption. Conversely, the avoidance of inertial resonance in our plasmacoustic metalayer is literally changing the game arena, by providing almost perfect sound absorption with a very small thickness over two frequency decades.

**Generalization to arbitrary reflection control**

Moving away from perfect absorption, the active plasmacoustic metalayer is also capable of controlling arbitrarily the incident sound wave, by manipulating the magnitude and phase of the reflection (Fig. 4). For this purpose, we employ a passive acoustic impedance resembling equation (4), which is modified into:

$$Z_{tg} = Z_c \frac{\beta Z_l + j Z_c \tan(k(l + x_0 + \Delta l))}{Z_c + j \beta Z_l \tan(k(l + x_0 + \Delta l))},  \quad (5)$$

where $\beta$ is a dimensionless scaling factor of the termination impedance $Z_l$, and the parameter $\Delta l$ relates to the virtual actuator thickness. If $\beta = 1$ and $\Delta l = 0$, the target impedance equals to passive $Z_{ac}$ and no active control is required. However, as $\beta$ changes, the target imitates the presence of a termination with impedance equal to $\beta Z_l$ placed right behind the metalayer. Lowering the value of $\beta$ leads to the softening of the virtual termination, thus, to higher sound absorption and lower reflection coefficient. When $\beta = Z_c/Z_l$, the target impedance equals the characteristic air impedance, yielding the full absorption case discussed previously. Figure 4a reports three experimental measurements demonstrating broadband sound control with different reflection levels, corresponding to 90, 70, and 40%, while the reflection phase remains close to zero.

When the parameter $\Delta l$ differs from zero, the sound wave reflects with a delay that corresponds to an additional propagation path of length $2\Delta l$. Therefore, besides only varying the reflection magnitude, we can artificially increase the size of the system, inducing delays that would normally occur over lengths exceeding by far the actual size of the device (~3 cm with enclosure). Effectively, we expand or squeeze the space seen by the sound wave. The concept is demonstrated experimentally in Fig. 4b. For a fixed reflection magnitude of 40%, we have set $\Delta l$ to 3.7, 9.7, 14.7 cm, which corresponds to reflection delays of 0.22, 0.56, 0.85 ms. These values are consistent with the slopes measured on the linear reflection phases curves. These results suggest that active plasmacoustic metalayers may be used to target on-demand absorption spectra, for example targeting good absorption in a certain frequency range and good reflection in another, with a full control over the reflection phases. In summary, the broadband sound manipulation offered by controlled subwavelength plasmacoustic metalayers is versatile, and offers a new playground for programmable acoustic metasurfaces.

## Discussion

In this paper, we have presented an active plasmacoustic metalayer capable of manipulating audible sound waves over two decades of frequencies, efficiently manipulating wavelengths as large as 17 m despite a total thickness of only a few mm. By showcasing broadband perfect sound absorption and several examples of magnitude and phase control of the reflected signal, we have experimentally established the non-resonant behavior and the tunability of the concept. A remarkable feature is that the plasmacoustic metalayer does not introduce any physical discontinuity within the fluid medium, due to the absence of a membrane. Instead, it produces controllable monopolar and dipolar acoustic sources directly in the fluid, without the need for inertial mechanical elements, thereby circumventing the drawbacks of standard passive and active sound control strategies.

Its simplicity of construction and versatility in terms of control and target functionality opens up the way for new directions in active sound control and acoustic metamaterial research, in which efficient

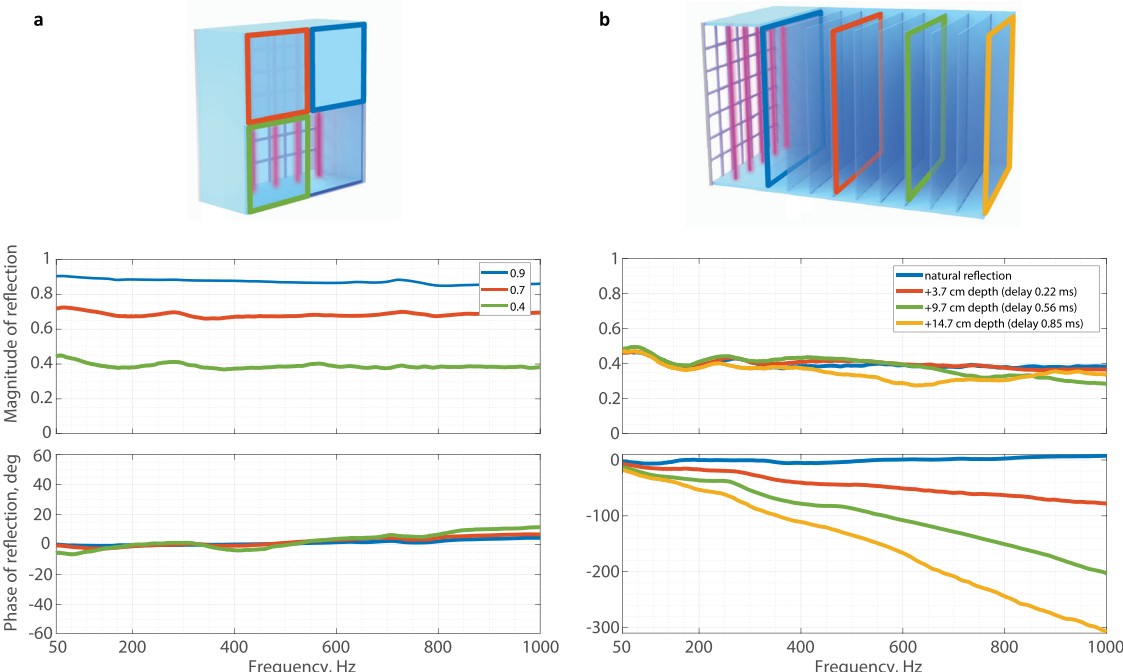

**Fig. 4 | Demonstration of a tunable mirror based on a plasmacoustic metalayer.** By adjusting the target impedance in equation (5), we can create a plasmacoustic interface with variable magnitude and phase of sound reflection. **a** Different magnitudes of sound reflection coefficient are demonstrated. The reflectance is set at 90, 70, and 40%, keeping the phase close to zero. **b** At a fixed reflection level, we can control the linear roll-off of the reflection phase, artificially introducing a constant reflection time delay. Delays of 0.22, 0.56, 0.85 ms correspond to virtual elongations of the device by 3.7, 9.7, 14.7 cm, respectively.

non-Hermitian, time-dependent and reconfigurable unit cells are much sought. The plasmacoustic metalayer is transparent to sound in the passive regime which is essentially different to most passive and active devices. Together with unprecedented bandwidth, plasmacoustic metalayers could be used as building blocks in a new generation of non-reciprocal, non-Hermitian and topological systems with unprecedented bandwidth[10,26,27], bringing them closer to real applications. The control law could be varied in time to break time-invariance, opening the door for the exploration of the acoustics of time-varying systems[28]. We stress that the shape of the electrodes and the active area can be modified at will, without any impact on the low and medium frequency responses. This may find promising applications in complex source design[29], imaging[30,31], large metasurfaces and acoustic arrays[32], especially when a nonconventional shape of the active unit cell is required. Altogether, our work establishes active plasmacoustic metalayers as a promising broadband approach for deep-subwavelength control of sound.

## Methods

### Derivation of control law

We consider an air-filled duct assuming plane wave propagation (for frequencies below plane waves cut-off), thus limiting the study to a one-dimensional problem. The duct is terminated by the plasmacoustic metalayer, with cross-section equal to the one of the duct. We can consider force **F** and heat release $H$ as point sources located at the same position with effective densities: $f = F_\omega/Sd$, $h = H_\omega/Sd$. This assumption is valid for various sizes of cross-sections $S$ since the distributions of force **F** and heat release $H$ are homogeneous along the metalayer area[23] and the inter-electrode distance $d$ with wires spacing (6 mm and 10 mm respectively) are sufficiently smaller than the wavelength. A simplified one-dimensional problem is schematically summarized in Fig. 2. Without any loss of generality, it can be assumed that the plasmacoustic layer is centered at the origin of the $x$ axis. As the air in the layer is weakly ionized, the energy mostly transfers through elastic interactions[22]. The wave equation can be derived from the system of linearized continuity,

Newton's, and energy conservation equations with heat and force sources, and the gas obeying adiabatic transformations[33]. These one-dimensional wave equations for acoustic pressure $p$ and particle velocity $v$ in the frequency domain are written as follows:

$$\left(\frac{\partial^2}{\partial x^2} + k^2\right)p = -\frac{j\omega}{C_P T_0}h + \frac{\partial f}{\partial x},$$
$$\left(\frac{\partial^2}{\partial x^2} + k^2\right)v = -\frac{1}{\rho_0 C_P T_0}\frac{\partial h}{\partial x} + \frac{j\omega}{\rho_0 c^2}f,$$
(6)

where $k = \omega/c$ is the wave number, $C_P$ is the heat capacity of air per unit mass, $T_0$ is the static ambient air temperature, $\rho_0$ is the static air density, $c$ is the speed of sound in the air. The external acoustic excitation emanates from the left relative to the plasmacoustic layer (negative $x$ values). Further, all acoustic pressures and velocities are considered for negative coordinates $x$ (where the measurement microphone is located). Using the principle of superposition, acoustic pressures and velocities generated only by either the heat source or the force source can be found independently from equation (6). The formulas read:

$$p_f(x) = \frac{d}{2}\left(e^{jkx} - Re^{jk(x-2l)}\right)f,$$
$$v_f(x) = -\frac{d}{2\rho_0 c}\left(e^{jkx} - Re^{jk(x-2l)}\right)f,$$
$$p_h(x) = \frac{dc}{2C_P T_0}\left(e^{jkx} + Re^{jk(x-2l)}\right)h,$$
$$v_h(x) = -\frac{d}{2\rho_0 C_P T_0}\left(e^{jkx} + Re^{jk(x-2l)}\right)h.$$
(7)

In equation (7), $R$ is the reflection coefficient from the back-enclosure wall ($x = l$) defined by $R = (Z_l - Z_c)/(Z_l + Z_c)$. The solutions of equation (6) are the sums $p_f + p_h$ and $v_f + v_h$ respectively. The microphone at position $-x_0$ senses $p_t(-x_0)$ which is then

considered as known. Using the relation $v_{ac}(x) = p_{ac}(x)/Z_{ac}(x)$ in (3) yields at coordinate $-x_0$:

$$v_t(-x_0) = v_f(-x_0) + v_h(-x_0) + \frac{p_t(-x_0) - p_f(-x_0) - p_h(-x_0)}{Z_{ac}(-x_0)}. \quad (8)$$

Taking into account that $v_t(-x_0) = p_t(-x_0)/Z_{tg}$ as we aim to realize a target impedance at position $-x_0$ and substituting all components from (7) in (8), one can obtain

$$\theta(k) = \frac{u_{AC}}{p_t(-x_0)} = \frac{1}{A}\left(1 - \frac{Z_{ac}(-x_0)}{Z_{tg}}\right) / \left(1 + \frac{Z_{ac}(-x_0)}{\rho c}\right),$$
$$A = \frac{d}{2}\left(f_0 + \frac{ch_0}{C_P T_0}\right)e^{-jkx_0} + \quad (9)$$
$$+ \frac{d}{2}\left(\frac{ch_0}{C_P T_0} - f_0\right)Re^{-jk(x_0 + 2l)}$$

In equation (9) $f_0 = f/u_{AC} = \frac{C}{S\mu_i}(2U_{DC} - U_0)$, $h_0 = h/u_{AC} = \frac{C}{Sd}(3U_{DC}^2 - 2U_{DC}U_0)$ are the parameters which depend on the actuator geometry and operating conditions and should be identified in advance. The transfer function $\theta(k)$ converts the pressure signal sensed in front of the actuator into an AC component of the electrical voltage signal $u_{AC}(k)$ which is further summed with $U_{DC}$ and applied to the electrodes. It is possible to adapt to different acoustic conditions, for example achieving partial or full absorption under any oblique incidence of sound (from normal to grazing incidence) by modifying the target impedance $Z_{tg}$ accordingly. However, before implementation on a digital platform, the transfer function should be expressed in terms of the Laplace variable $s = j\omega$. One must ensure that $\theta(s)$ is a stable and proper transfer function.

The theoretical model developed here allows investigating the contributions of monopolar and dipolar sources in the metalayer in case of perfect sound absorption. Supplementary Fig. 2 shows that, in the low frequency range, the pressure signal is mostly dominated by the monopolar $p_h$ source, whereas it is the dipolar $p_f$ source at higher frequencies. The crossover frequency depends on parameter $l$ of the model and the relative strength of the heat and force sources. Such complementarity of the main contributing sources allows broadband efficiency compared to conventional transducers which always represent a bandpass behavior, owing to their resonant nature.

### Experiment

The measurements under normal incidence are carried out in an impedance tube of length $L_{duct} = 1.1$ m and square cross-section $S_{duct} = 50 \times 50$ mm$^2$ which is schematically illustrated in Fig. 3a. A loudspeaker is mounted at the left termination of the duct and generates a swept-sine signal in the frequency range 20-2000 Hz with the incident amplitude of 1 Pa. The plasmacoustic metalayer closes the duct at the right termination. The distance $d = 6$ mm between the electrodes was chosen as a trade-off between satisfying the assumptions of the analytical model and ensuring the stability of the discharge. It is backed by a rectangular enclosure with depth $l = 15$ mm (with respect to the metalayer center $x = 0$) and cross-section $50 \times 50$ mm$^2$. One Piezotronic PCB 130D20 ICP microphone senses the total sound pressure in front of the metalayer 10 mm away at the left from its center. The control transfer function $\theta(s)$ from (9) with $Z_{tg} = \rho c$ is digitized and implemented on the real-time platform Speedgoat IO-334 which runs at 50 kHz frequency. As the controller cannot generate high voltages, it calculates $u_{AC}(s)$ component reduced by a factor 1000 for the output. Controller is then connected in series with a floating power supply that adds $U_{DC}$ (8 V to achieve 8 kV). The power supply output feeds an input of a TREK 615-10 high voltage AC/DC amplifier with amplification factor of 1000 that supplies the plasmacoustic layer. Two additional microphones are placed along the

duct, with inter-distance 50 mm, the closest microphone being at 35 cm from the collector grid. They are used to estimate the acoustic impedance and sound absorption coefficient according to ISO-10534-2 standard[34]. The microphones signals are processed with a Brüel & Kjaer Pulse Multichannel Sound and Vibration analyzer. Measurements are performed in the 20-2000 Hz frequency range since the theoretical assumption of an infinitely thin plasmacoustic layer is not valid at high frequencies. Moreover, controller delay and sampling rate also affect sound absorption performance at higher frequencies.

### Discharge parameters estimation

To estimate the behavior of the plasmacoustic metalayer with analytical model and set up the control transfer function $\theta(s)$, several parameters related to the corona discharge process should be estimated. They include the effective mobility of positive ions $\mu_i$, the offset voltage $U_{DC}$, the critical voltage $U_0$, and constant $C$. The value of $\mu_i$ is taken from[35], corresponding to a relative humidity 50–55%, as it was observed during the measurements. The other parameters are estimated from experimentally-obtained voltage-current curve (Supplementary Fig. 1). The total electrical current in the metalayer is measured at voltages up to 10 kV. Based on this, $U_{DC} = 8.0$ kV is chosen as the bias voltage as it corresponds to the center of the operating range. $U_{DC}$ should be high enough to keep the total voltage for a metalayer in a corona discharge regime, otherwise, the response of the metalayer can be nonlinear after the application of the alternating voltage due to the lack of ions. The metalayer prototype consumes electrical power of 2.5 W to maintain the ionization at 8 kV bias voltage. The experimental data is fitted by the Townsend's formula around 8 kV (red line in Supplementary Fig. 1) that gives the estimations for $C$ and $U_0$. The values of all parameters used to build the control transfer function for the experiment are listed in Supplementary Table 1. It should be noted that, in the case of a significant change in environmental parameters such as air humidity, ambient pressure and temperature, the ionization process differs due to the change of gas mixture and energy of molecules. This can affect the shape of the discharge voltage-current characteristics, thus, the dynamics of the plasmacoustic metalayer for sound control, and cause unstable behavior of the control system. Therefore, these parameters should be re-estimated. This can be potentially performed automatically.

### Data availability

The measurement data that support the figures and other findings within this paper are openly available at https://zenodo.org/record/7867877.

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

## Author contributions

All authors contributed to the development of the concept and writing the manuscript. S.S. carried out theoretical and experimental work. H.L. and R.F. supervised the project and proposed the sound manipulation scenarios.

## Competing interests

The authors declare no competing interests.
