## [Peer Review File · Nature Communications]

Ultrabroadband sound control with deep-subwavelength plasmacoustic metalayersReviewer #1 (Remarks to the Author):

In this manuscript, the authors present a new active sound control solution. Therein, they build a very thin plasmacoustic layer [about ($\sim\lambda/1000$)] which physically plays the role of active monopolar and dipolar sources once modulated voltage is applied. To exemplify the potential of their active acoustic device, the authors present two scenarios, i.e., perfect sound absorption in a broad frequency interval (20Hz \sim 2kHz) and arbitrary reflection control in both magnitude and phase. These two cases indeed show the great tunability of this active setup. In this regard, this work could benefit the sound control community. Before I give the final decision, I would like the authors to address the following comments.

The plasmacoustic layer functions as the combination of monopolar and dipolar sources, thus a quantitative analysis and discussion of the contribution of these two sources in the presented applications should be included.

In the derivation of the control law, the authors claim the heat release is homogeneous among the cross-section area. So does this assumption always hold when the section turns larger, if not, please clarify the limitation.

Also, is there any specific limitation on the distance between two electrodes, please clarify this.

How about the temperature change in the emitter of the metalayers, for example in Fig. 3(c), does it suffer from temperature disturbance? Some discussion of the robustness of this active setup is needed.

Reviewer #3 (Remarks to the Author):

This paper presents an interesting application of air plasma in the development of deep subwavelength acoustic absorbers. The revised manuscript has added clarifications on the key issues raised by the reviewers in the previous submission to Nature Physics. However, the reviewer thinks that it has limitations in the following areas that the authors can address in the revision.

First, the idea of actively modulating a thin layer of air plasma seems to be novel for achieving perfect absorption. As Reviewer #1 pointed out, the theoretical framework has been developed and published by the same authors - those publication(s) should be cited in this manuscript. By measuring the pressure instead of velocity, the number of sensors is deduced from two to one, which has some advantage. Whether the new development is incremental is a question.

Second, beyond the theoretical development, the developed devices have limited practicability due to its dependence on incident angle and humidity, safety hazard (8,000 Volt), and high energy consumption (2.5 W for an area of $5 \times 5 \text{ CM}^2$).

Third and most importantly, it is questionable to claim the presented device can achieve perfect sound absorption in a very broad frequency range (two decades from 20 Hz to 2 kHz). It is more accurate to describe the device as one that is tunable in a broad frequency range. The resonator-based metamaterials (e.g. the membrane devices by Ping Sheng) can achieve perfect narrow-band absorption, but they can be tuned, too (not actively though). A test that can differentiate a device that is truly broadband from one that is narrow-band but is tunable is to change the incident sound from a puretone to a broadband sound - can the active modulating system still work?

NCOMMS-23-0153: Reply to reviewers

Reviewer #1 still has concerns on the limitation of the approach demonstrated here, and asks for differentiation of monopolar and dipolar sources contribution in the plasmacoustic layer. The newly invited Reviewer #3 expresses concern regarding the conceptual novelty of your work and recommends putting your work into context to previously published similar work of yours. Moreover, Reviewer #3 questions the practicality of the device in light of the high operating voltage and energy consumption needed for operation.

In light of these comments, we cannot accept the manuscript for publication, but would be interested in considering a revised version that addresses these serious concerns.

First, we would like to thank the two reviewers for their evaluation of our work. We have provided in the following a point-by-point detailed reply to their comments, and the changes made to the manuscript are given in blue after the corresponding comment. We hope that our modifications add clarity and improved the manuscript, and that this revised version will meet the reviewers' expectations.

Reviewer #1

In this manuscript, the authors present a new active sound control solution. Therein, they build a very thin plasmacoustic layer [about ($\sim\lambda/1000$)] which physically plays the role of active monopolar and dipolar sources once modulated voltage is applied. To exemplify the potential of their active acoustic device, the authors present two scenarios, i.e., perfect sound absorption in a broad frequency interval (20Hz ~ 2kHz) and arbitrary reflection control in both magnitude and phase. These two cases indeed show the great tunability of this active setup. In this regard, this work could benefit the sound control community. Before I give the final decision, I would like the authors to address the following comments.

1. *The plasmacoustic layer functions as the combination of monopolar and dipolar sources, thus a quantitative analysis and discussion of the contribution of these two sources in the presented applications should be included.*

Reply: We thank the reviewer for pinpointing the important role and the complementarity of the two equivalent sound sources, as it explains the broadbandness of the proposed concept in terms of sound manipulation (absorption/reflection), and its advantages against membrane-based concepts. We have added more information about the monopolar and dipolar sources properly citing our previous work. We have also added a detailed breakdown of monopolar/dipolar contributions in the Methods section, supporting this with supplementary figures:

- Line 76: “Derivation of equations (1-2) and their contributions to the total generated pressure in free field are discussed in the previous work [23].”

- Line 99: “See the derivation and breakdown of their contributions to the final system response in Methods.”
- Line 220 (Methods section): “The theoretical model developed here allows investigating the contributions of monopolar and dipolar sources in the metalayer in case of perfect sound absorption. Figure 2 in supplementary material shows that, in the low frequency range, the pressure signal is mostly dominated by the monopolar p_h source, whereas it is the dipolar p_f source at higher frequencies. The crossover frequency depends on parameter l of the model and the relative strength of the heat and force sources. Such complementarity of the main contributing sources allows broadband efficiency compared to conventional transducers which always represent a bandpass behaviour, owing to their resonant nature.”

Supplementary figure 2. Magnitudes of sound pressure signals, derived from equation 7, generated by a plasmacoustic metalayer at the microphone position, in response to an incident sound source of amplitude 1 Pa.

2. In the derivation of the control law, the authors claim the heat release is homogeneous among the cross-section area. So does this assumption always hold when the section turns larger, if not, please clarify the limitation.

Reply: Thank you for this very pertinent question. Indeed, assuming homogeneity along the section of a plasmacoustic metalayer allows relying on global quantities that are directly accessible to measurements: total electrical current and voltage. This is actually true as long as the parallel wire portions of the high voltage electrode are close to each other (distant by 10 mm in our case). With such a geometry, the transducer can be considered as a source of plane waves up to ~ 10 kHz. Therefore, the same behaviour is expected independent of the cross-section of the transducer. We have added a sentence in the Methods section clarifying this limitation of plane wave generation. This also replies the following question 3.

- Line 192: “This assumption is valid for various sizes of cross-sections S since the distributions of force F and heat release H are homogeneous along the metalayer area [23] and the inter-electrode distance d with wires spacing (6 mm and 10 mm respectively) are sufficiently smaller than the wavelength.”

3. *Also, is there any specific limitation on the distance between two electrodes, please clarify this.*

Reply: Indeed, the interelectrode distance influences at least two aspects:

- The electric field magnitude depends on the interelectrode distance and the applied voltage difference. If the distance is larger than the one considered in this work (6 mm), the device would require higher operating voltages to generate the same electric field, and consequently, the generated acoustic signal. On the other hand, if the distance between the electrodes is much smaller than the one used in this work, any micro-roughness over the electrodes surface is likely to generate locally strong electric field, which leads to an earlier breakdown and lowers the discharge stability [Moreau, 2007];
- The distance between the electrodes should be smaller than the wavelength since the analytical model assumes that the discharge generates acoustic signals only at coordinate $x=0$. This assumption is satisfied in the considered frequency range but would not allow operating at much higher frequencies, for example, above 10 kHz.

In this work, the inter-electrode distance of 6 mm was determined as a trade-off between these two most influential properties. We have added a text in the section describing the experiment:

- line 230: “The distance between the electrodes $d=6$ mm was chosen as a trade-off between satisfying the assumptions of the analytical model and ensuring the stability of the discharge.”

4. *How about the temperature change in the emitter of the metalayers, for example in Fig. 3(c), does it suffer from temperature disturbance? Some discussion of the robustness of this active setup is needed.*

Reply: The metallic emitter within the metalayer is made of 0.1 mm in diameter nichrome wire. This wire has a resistivity ~ 100 Ohm/m. Approximately 1 m of the wire is used to build the device (50 cm exposed for ionization and some parts inside the frame of the device). Since the metalayer consumes well below 1 mA of current (supplementary Figure 1), the power loss inside the cable does not exceed 0.1 mW. Therefore, the emitter does not heat due to conduction of current. A heat release

happens due to inelastic collisions and ionisation process in the air around the emitter that can lead to an increase of ambient temperature by a few degrees if the device is placed in a small confined volume. This can result in a change of the discharge dynamics and change of effective parameters used to establish the theoretical model, therefore requiring to update the parameters accordingly. Thank you for this comment, we included a remark regarding the temperature in the Methods section.

- Line 258: “It should be noted that, in the case of a significant change in environmental parameters such as air humidity, ambient pressure and temperature, the ionisation process differs due to the change of gas mixture and energy of molecules. This can affect the shape of the discharge voltage-current characteristics, thus, the dynamics of the plasmacoustic metalayer for sound control, and cause unstable behaviour of the control system. Therefore, these parameters should be re-estimated. This can be potentially performed automatically.”

Regarding the results presented in Figure 3c, the deviation from ideal absorption ($\alpha=1$) is not linked to such temperature disturbances, but can be explained by several reasons. At very low frequencies (below 100 Hz), the spacing between the measurement microphones (50 mm) may not be enough to precisely resolve sound wave lengths of several metres. In the high frequency region (above 1500 Hz), the theoretical model assumptions of an infinitely thin plasma metalayer does not hold anymore (thickness of the order of one-tenth of the wavelength) and it might introduce greater error in the control than at lower frequencies. In addition, we use a real-time controller which has a certain time delay in the range of 20 microseconds due to sampling which introduces a mismatch between the target impedance and the one achieved at high frequencies.

Reviewer #3

This paper presents an interesting application of air plasma in the development of deep subwavelength acoustic absorbers. The revised manuscript has added clarifications on the key issues raised by the reviewers in the previous submission to Nature Physics. However, the reviewer thinks that it has limitations in the following areas that the authors can address in the revision.

1. *First, the idea of actively modulating a thin layer of air plasma seems to be novel for achieving perfect absorption. As Reviewer #1 pointed out, the theoretical framework has been developed and published by the same authors - those publication(s) should be cited in this manuscript. By measuring the pressure instead of velocity, the number of sensors is deduced from two to one, which has some advantage. Whether the new development is incremental is a question.*

Reply: We thank the reviewer for questioning the difference between this manuscript and our previously reported works (which has been already cited in the manuscript as [23]) and [Sergeev et al, 2022], allowing us formulating an argumentation on the novelty presented here. Indeed, in a recent paper [Sergeev et al, 2022] we demonstrated the use of a similarly constructed corona discharge actuator in an active sound absorption concept. In this work, the transducer was employed in two alternative configurations:

- in combination with a porous layer (wire mesh), where the transducer aimed at assigning pressure minimization behind the wiremesh relying on adaptive FxLMS algorithm;
- without the porous layer, but achieving a prescribed target acoustic impedance, through velocity feedback, where both the velocity (with a microphone pair) and the total pressure are measured, and the control is achieved by minimising the difference between the measured velocity and the targeted one (total pressure/ Z_{target}), therefore where the transducer technology does not account (it can be done with any kind of loudspeaker instead).

The first setup is definitely different, as it relies on the passive absorption through the wire mesh. However, although the second setup might appear similar to the one reported in the present manuscript at first glance, it is conceptually - and effectively performance-wise - different on multiple aspects.

This second setting is not only much bulkier than the one proposed due to use of the microphone pair (which we acknowledge would only represent an incremental development), but it uses the plasma transducer as a black box without taking any advantage of its physical principle. Thus, any loudspeaker could be used for such type of impedance matching control through minimization of velocity difference. More specifically, the control law does not require any model of the transducer (and it is simply a feedback gain) to achieve such impedance matching. Moreover, the absorption performance achieved under normal incidence with such a method is incomparably lower than in the current work.

What makes the proposed concept unique is that it essentially relies on the physical model of the transducer. In our opinion, the huge step presented here relies on taking advantage of the physical processes inside the corona discharge to interact with an exogenous sound field: heat transfer and generation of electromechanical force and leveraging them into a broadband sound absorbing device.

We have added a proper citation of our previous works:

- line 76: “These two phenomena can be included as sound source terms in the acoustic wave equation. Derivation of equations (1-2) and their contributions to the total- generated pressure in free field are discussed in the previous work [23]. In [24], an attempt was made to control the Corona Discharge transducer was used as

a generic transducer to absorb sound, without benefiting from the physics of the corona discharge. However, equations (1-2) are frequency-independent, which suggests that they can form the basis for an active actuator that can be controlled over a potentially broad frequency range, as we now demonstrate.”

2. *Second, beyond the theoretical development, the developed devices have limited practicability due to its dependence on incident angle and humidity, safety hazard (8,000 Volt), and high energy consumption (2.5 W for an area of 5x5 cm²).*

The impedance does not depend on angle, but absorption and reflection do. To cope with angle-dependent refraction, one can use the generalised Snell-Descartes law as was proposed in the previous reply to reviewers (please, refer also to H. Esfahlani et al, Physical Review B, 94, 014302 (2016)). In case of random sound incidence the optimal impedance can be chosen that suits the best for a given sound manipulation (absorption, steered reflection).

The changes in environmental parameters such as humidity indeed affect the performance of the plasmacoustic metalayer. A short discussion on the issue and mitigation of it is proposed in the response to the comment 4 of Reviewer 1.

The use of high voltage does not introduce any issues if the electrical safety rules are respected. High voltages are used in consumer electrical appliances such as air purifiers and laser printers. The plasmacoustic device has a grounded collector grid and the high voltage part is out of reach inside the enclosure, so it can be designed in a safe manner.

The energy consumption is indeed higher than from a conventional loudspeaker. However, the benefits in sound control the device provides can help reducing the weight of passive sound materials used and by this increase energy efficiency well beyond the consumption of plasmacoustic system.

3. *Third and most importantly, it is questionable to claim the presented device can achieve perfect sound absorption in a very broad frequency range (two decades from 20 Hz to 2 kHz). It is more accurate to describe the device as one that is tunable in a broad frequency range. The resonator-based metamaterials (e.g. the membrane devices by Ping Sheng) can achieve perfect narrow-band absorption, but they can be tuned, too (not actively though). A test that can differentiate a device that is truly broadband from one that is narrow-band but is tunable is to change the incident sound from a pure tone to a broadband sound - can the active modulating system still work?*

Reply: We don't agree that the presented device is only tunable in the values of achievable absorption/reflection. It is essentially broadband (over frequencies) as

illustrated in the different measurements (exceeding 95% absorption between 20 H and 2 kHz, or constant reflection coefficients with various values of phase). To alleviate any misunderstanding, we shall precise that all the measurements were done with a chirp acoustic signal exciting the impedance tube, without modifying the control of the active device during the measurement. The same can also be achieved with any other excitation (stationary broadband/band-limited noise, modulated noises, impulsions, etc.), which contains frequencies in the same range up to 2 kHz. In the following figure, an alternative sound absorption measurement (in blue) is performed with the noise source being a broadband noise instead of a sinusoidal sweep as provided in the main paper (red curve). The achieved sound absorption does not significantly differ from the measurement presented in the paper, thus proving the broadbandness of the concept.

Figure showing the same measured performance of the absorber whether the external excitation is presented as a broadband noise or a sinusoidal sweep.

Reviewer #1 (Remarks to the Author):

The revised version is satisfactory. I support its publication.

Reviewer #3 (Remarks to the Author):

The authors have addressed all the concerns and questions raised by the reviewers.